# Halobacteria-Based Biofertilizers: A Promising Alternative for Enhancing Soil Fertility and Crop Productivity under Biotic and Abiotic Stresses—A Review

**DOI:** 10.3390/microorganisms11051248

**Published:** 2023-05-09

**Authors:** Fatma Masmoudi, Mohammed Alsafran, Hareb AL Jabri, Hoda Hosseini, Mohammed Trigui, Sami Sayadi, Slim Tounsi, Imen Saadaoui

**Affiliations:** 1Biotechnology Program, Center for Sustainable Development, College of Art and Sciences, Qatar University, Doha P.O. Box 2713, Qatar; fatma.masmoudi@qu.edu.qa (F.M.); h.aljabri@qu.edu.qa (H.A.J.); hoda.hosseini@qu.edu.qa (H.H.); ssayadi@qu.edu.qa (S.S.); 2Central Laboratories Unit (CLU), Office of VP for Research & Graduate Studies, Qatar University, Doha P.O. Box 2713, Qatar; m.alsafran@qu.edu.qa; 3Agricultural Research Station (ARS), Office of VP for Research and Graduate Studies, Qatar University, Doha P.O. Box 2713, Qatar; 4Department of Biological and Environmental Sciences, College of Art and Sciences, Qatar University, Doha P.O. Box 2713, Qatar; 5Laboratory of Environmental Sciences and Sustainable Development (LASED), Sfax Preparatory Engineering Institute, University of Sfax, Sfax 3018, Tunisia; mohamed.trigui@ipeis.usf.tn; 6Laboratory of Biopesticides (LBPES), Center of Biotechnology of Sfax, University of Sfax, Sfax 3038, Tunisia; slim.tounsi@cbs.rnrt.tn

**Keywords:** abiotic stress, biofertilizer, biotic stress, halobacteria, soil fertility, sustainable agriculture

## Abstract

Abiotic and biotic stresses such as salt stress and fungal infections significantly affect plant growth and productivity, leading to reduced crop yield. Traditional methods of managing stress factors, such as developing resistant varieties, chemical fertilizers, and pesticides, have shown limited success in the presence of combined biotic and abiotic stress factors. Halotolerant bacteria found in saline environments have potential as plant promoters under stressful conditions. These microorganisms produce bioactive molecules and plant growth regulators, making them a promising agent for enhancing soil fertility, improving plant resistance to adversities, and increasing crop production. This review highlights the capability of plant-growth-promoting halobacteria (PGPH) to stimulate plant growth in non-saline conditions, strengthen plant tolerance and resistance to biotic and abiotic stressors, and sustain soil fertility. The major attempted points are: (i) the various abiotic and biotic challenges that limit agriculture sustainability and food safety, (ii) the mechanisms employed by PGPH to promote plant tolerance and resistance to both biotic and abiotic stressors, (iii) the important role played by PGPH in the recovery and remediation of agricultural affected soils, and (iv) the concerns and limitations of using PGHB as an innovative approach to boost crop production and food security.

## 1. Introduction

Malnutrition is a global problem that affected approximately 811 million people in 2020. The growing world population, estimated to reach 9.7 billion by 2050 [1], will accentuate malnutrition problems and cause a noticeable global deterioration in food security, particularly in developing countries. To meet this increasing demand, food production needs to increase by approximately 70% by the year 2050 [2]. However, intensive agricultural practices have various negative effects on soil. It is the main reason for physical degradation caused by agricultural machinery, drainage, and dewatering. The overuse of manure and fertilizers can lead to organic and inorganic pollution, which can result in soil salinization, acidification, and chemical degradation. Moreover, synthetic fertilizers, pesticides, and monoculture can negatively affect soil biodiversity and ecosystem functioning, leading to a biological degradation of soil organic matter, which is essential for soil structure, water-holding capacity, and nutrient cycling [3].

Additionally, intensive agriculture is responsible for approximately 92% of freshwater consumption, raising several concerns about the use of water resources [4]. Irrigation with saline water accentuates the soil salinization process and affects yield production.

In other words, a growing demand for agricultural products has been observed, while the amount of cultivated land, the soil structure, and availability of natural resources has been steadily decreasing [5,6,7,8]. Thus, global agriculture’s capacity will no longer be able to sustain the increasing demand of nutriment supply without compromising environmental integrity and human health. To address these challenges, biotechnological tools such as microbial bioinoculants have been developed to provide eco-friendly and sustainable solutions to enhance food production. Among these tools are halotolerant bacteria, which are considered nature-based solutions [9]. Plant-growth-promoting halobacteria (PGPH) have been found to be highly effective in promoting plant growth in non-saline conditions and improving soil health, particularly in saline and arid environments [10]. These bioinoculants have the ability to adapt to high salt concentrations and aid plants in coping with the stress of saline soils [8]. Furthermore, using bioinoculants can lead to a reduction in the use of chemical fertilizers and pesticides, making them an eco-friendly alternative.

Several previous reviews highlighted the important role played by PGPH in plant growth promotion and mitigating plant drought and salt stresses, and most of these studies have focused solely on this aspect [11,12,13,14]. Only a few reports focused on the potential of PGPH to play other roles, such as in biocontrol and the recovery of contaminated soils [14,15].

Interestingly, to the best of our knowledge, no previous review article summarizes in a single paper the various roles played by PGPH in protecting and increasing crop productivity under biotic and biotic stresses, recovering degraded lands, and sustaining agricultural products. Our review article is the first to explore the multitasking ability of halotolerant bacteria isolated from saline environments and halophyte rhizospheres, which contribute not only to alleviating abiotic stresses but also to improving plant protection against biotic diseases, promoting plant growth, and aiding in the recovery of affected soils. It also discusses the challenges facing conventional agriculture, particularly biotic (fungal attack) and abiotic stresses (salt stress, drought stress, etc.). The review also highlights the direct and indirect pathways used by PGPH as bioinoculants to improve crop tolerance to abiotic stresses and fight against phytopathogenic attacks. Finally, it highlights PGPH’s limits in agriculture and presents the authors’ recommendation for future research and application of halotolerant bacteria as biofertilizers.

## 2. Challenges Limiting the Sustainability of Conventional Agriculture

A variety of biotic and abiotic stresses can negatively affect crop yield and food safety when they are applied to agricultural crops. The biotic stressors include but are not limited to insect pests, phytopathogenic fungi, bacteria, and viruses, while abiotic stresses include soil salinization, drought, heavy metals, temperature variations, nutrient availability, mineral deficiency, etc. All these stresses have a huge negative impact on the fertility of agricultural lands and reduce average yields by more than 50% throughout the world [16]. Thus, they can cause serious agricultural issues and critical environmental problems, presenting a real ecological and socioeconomic threat to sustainable development (Figure 1).

### 2.1. Soil Salinization

A widespread agricultural issue that affects many different types of land surfaces worldwide is soil salinization. Lands affected by salt are those whose electrical conductivity of the saturation extract (ECe) is more than 4 dS m^−1^ [17]. ECe is a measure of soil salinity caused by the presence of different ions such as sodium (Na+), calcium (Ca^2+^), magnesium (Mg^2+^), sulfate (SO_4_^2−^), chloride (Cl^−^), and bicarbonate (HCO^3−^) without providing a direct measure for specific ions. Amongst these ions, sodium and chloride are considered the most toxic due to their critical role in soil structure deterioration and plant toxicity [18]. Thus, defining the nature of the affected soil requires more than the measure of ECe. It is necessary to determine soil sodicity through the measure of two parameters:the concentration of Na^+^ ions relative to Ca^2+^ and Mg^2+^ ions, expressed in the sodium adsorption ratio (SAR) and the determination of exchangeable sodium percentage (ESP).In fact, soils are considered sodic when SAR and ESP are ≥13 and 15, respectively [17].

Primary salinity is a naturally occurring element of the environment that is created by geological, hydrological, and pedological processes. However, improper anthropogenic activities have exacerbated the soil salinization process (secondary salinity) [19]. Irrigation methods with poor quality of water alongside inefficient drainage systems are recognized as the main cause of secondary salinity. Irrigation with inadequate quality and quantity of water increases leakage rising and sea water intrusion in the over-exploited groundwater [19,20]. Due to salt migration in the top and lower soil layers, deforestation, clearance of vegetation, and changes in land use are also regarded as key contributors to land salinization and alkalization [21]. Additionally, modern intensive agricultural methods and farming systems result in the production of wastewater and effluent with high salt concentrations, as well as chemically contaminated soil that causes soil salinization, mostly in closed or semi-closed systems (greenhouses) [22]. Further, climatic factors such as low rainfall and high temperature play an important role in the pedogenesis of agricultural soils and may accentuate soil salinization. Such climatic factors exacerbate the accumulation of soluble salts such as chloride (Cl^−^), sodium (Na^+^), magnesium (Mg^2+^), potassium (K^+^), bicarbonate (HCO^3−^), sulfate (SO_4_^2−^), carbonate (CO_3_^2−^), and calcium (Ca^2+^) and cause the formation of salt-affected soils, especially in arid and semiarid areas, where evaporation exceeds precipitation [11]. All of these constituents significantly reduce land fertility, raise salt concentrations toward plant roots, and perturb plant metabolism and soil structure, which directly affects food security and economy at several levels: local, regional, and even global levels [23].

Salinity is recognized as the main cause of desertification [24]. Approximately 20% of the irrigated lands around the world are affected by salinity and sodicity problems [25]. In fact, the growing salinity, particularly in arid and semiarid irrigated regions, results in the loss of roughly 10 million hectares of agricultural land globally and around 1 million hectares in the European Union (mostly in the Mediterranean countries) each year [23,26]. Salt accumulation causes the soil structure’s degradation and deflocculation, increases osmotic pressure, and reduces water potential. Salinization disrupts ionic homeostasis and osmotic imbalance, which lead to physiological drought, nutrient imbalance, ion toxicity, increases in reactive oxygen species (ROS) level, production of stress ethylene, and, subsequently, the senescence and early death of the plant. Moreover, in saline soils, microbial diversities and densities are negatively affected, which alters soil fertility and productivity [27]. Understanding these modulations can help to manage the continuous propagation of global issueusing two major approaches: modifying the plant to be suitable for the environment or modifying the environment to be suitable for the plant.

### 2.2. Phytopathogenic Disease

Phytopathogenic microorganisms (bacteria, fungi, viruses, etc.) are considered the most harmful biotic factors to a wide range of plant crops. When accompanying other biotic and/or abiotic factors, they can cause crucial agricultural economic losses and environmental issues worldwide [28,29]. Fungi and bacteria are mainly responsible for plant infections. They affect all types of plants, colonize all plant tissues, and cause different symptoms including blasts, spots, blights, rusts, cankers, tissue rots, blights, and mildews, among others. They are also responsible for hormone imbalances and contribute to plant growth disruption, root branching, stunting, leaf epinasty, etc. [28,29,30]. Plant diseases, caused by phytopathogenic microorganisms and animal pests, are increasing yearly and are estimated to cross 38.2% and 36.5% of total yield losses in rice and potatoes, respectively [31,32]. Although bacterial diseases cause significant economic losses in food production, reaching 1 billion USD yearly worldwide [33], fungal phytopathogens represent the greatest threat to plant health. Plant diseases caused by phytopathogenic microorganisms and animal pests are increasing yearly and are estimated to exceed 38.2% and 36.5% of total yield losses in rice and potatoes, respectively [31,32]. Yield loss could occur pre-harvest as well as in post-harvest processes and could lead to annual economic losses exceeding 200 billion USD [34].

Numerous fungi are responsible for various diseases in different crop fields and plants. Sometimes, their symptoms are similar and, therefore, confusing. In addition to their high infection potentialities, fungi liberate important levels of highly toxic, hallucinogenic, and carcinogenic substances. For instance, Aspergillus species are known to produce aflatoxin, which is not only highly toxic but also considered a very carcinogenic substance [35]. Phytopathogenic fungi attack different organs of the plant host pre- or post-harvest. Some fungal species infect the external or aerial organs of the plant and their symptoms are plainly visible and noticeable. Unfortunately, most other fungi are soil-borne phytopathogens. They attack the root system and initiate the infection underground, making the preliminary symptoms difficult to detect.

Fungi usually infect plants and use them as a food source. Depending on their strategies of nutritional acquisition, plant pathogenic fungi are clustered into three different groups: biotrophs, necrotrophs, and hemibiotrophs [36,37]. Biotrophic fungi survive on living cells and use them as a source of nutrients. They infect the plant and absorb nutrients directly from living host tissues without causing programmed cell death (PCD). Biotrophic fungi can cause hyperplasia or hypertrophy of the affected organs such as pustules, smuts, and patches of damaged tissue. Powdery mildew fungus and rust fungi are examples of biotrophs with a limited host range [38]. Before infection and colonization, necrotrophs modify host cell plasma membranes and produce PCD, or they may subsequently obtain nutrients from dead cells. Root rots, trunk rots, trunk infections, post-harvest rots, and tracheomycosis are illnesses linked to necrotrophs. Necrotrophs cause bark lesions and xylem vascular damage [39]. Hemibiotroph infections start like biotroph infections, requiring living cells to grow. Then, after their development, they adopt necrotrophic pathogen traits and kill their host to feed on the dead tissues. They cause leaf damage and spot diseases [37,39].

### 2.3. Plant Sensitivity

Plants are subjected to complex interactions and combinations of stress elements because environmental conditions are always changing. The way that different species and even variations of plants react to stress differs. Stresses that are biotic and/or abiotic disrupt the physiological metabolism of plants from germination to maturity, which reduces plant growth and crop yield. Such stresses are responsible for the augmentation in ethylene production, which is a stress-signaling molecule leading to plant senescence and early death [40]. Additionally, stressing plants can impede their growth, hinder seed germination and growth, disrupt nutrient intake, and change the structure and function of the enzymes involved in nucleic acid metabolism. Thus, it affects protein metabolism and hormonal balance [19,41]. Stresses can also result in plants’ fluidity alteration, phospholipid membrane degradation, and generation of damage-associated molecular patterns (DAMPs) to decrease the stress’ damaging effects [42,43]. 

To survive in harsh environments, plants have evolved several mechanisms for the early detection of environmental variations and complex stress conditions. This helps plants to minimize damage and conserve their growth, production, and multiplication. Timely perception is considered a crucial step in plant defense in order to react efficiently [44]. When a stressor is detected, intricate defensive signaling cascades that are specific to the type of stress are activated. For example, plants may activate particular ion channels and kinase cascades, reprogram the genetic code, produce reactive oxygen species (ROS), or release a variety of phytohormones to boost their tolerance to unfavorable environments. (abscisic acid (ABA), salicylic acid (SA), jasmonic acid (JA), etc.), and the accumulation of ethylene (ET) [44]. However, in fields, plants are often exposed to a combination of biotic and abiotic stresses. Combinations of stresses can be categorized based on the number of interacting factors: a single stress factor, multiple stress factors occurring at different periods, and combined stress factors occurring simultaneously [45]. According to several studies, plants react complexly and differently to combined biotic and abiotic stressors, resulting in tolerance or susceptibility [46]. Abiotic stress generally makes plants more vulnerable to hemibiotrophic or necrotrophic bacteria and less vulnerable to biotrophic diseases. According to Feng et al. [47], salt stress causes cell wall damage, which disrupts intracellular defense signaling and leads to pathogen invasion. However, many other reports documented the improvement of plant resistance during combined biotic and abiotic stresses depending on the stress type. According to Bai et al. [48], the occurrence of high salt stress concurrent with powdery mildew attack restricts pathogen growth in tomato plants due to the osmotic and the ion toxicity effects exerted by high salinity levels.

## 3. Increasing the Adaptability of Plants to Stressors Using PGPH

Numerous studies have been conducted to create plans of action and solutions to reduce the harmful effects of biotic and/or abiotic stressors on plants. These techniques include breeding, genetic engineering methods, pesticides, organic amendments, irrigation and drainage techniques, and agriculture management practices. However, all these efforts have limited success due to the large diversity of stress factors, the increasing problem of water scarcity, and the rising cost of energy resources [11,41].

Pesticides are widely used to decrease the impact of these problems. However, several concerns have been raised over the systematic and extensive use of these synthetic products in agriculture. Such chemicals could be highly toxic, and their residuals alter the environmental balance and affect human health. Additionally, the emergence of pathogens resistant to chemical pesticides decreased the efficacy of these products and made the syntheses of new chemicals more difficult and costly [49]. Moreover, chemical pesticides fail to protect and increase yields when plants are exposed to harsh abiotic conditions, such as soil salinization.

To remedy this critical situation, there is an increasing need to develop sustainable strategies of high efficiency, to restore agricultural soil fertility, and of high plasticity, to be applied in vast geographic areas. Biopesticides derived from natural substances such as plants and microorganisms are promising alternatives to chemical products [50]. During the last decade, the number of commercial biopesticides significantly increased, constituting around 20% of phytosanitary products registered in the USA [30,51].

### 3.1. Mechanisms of Bacteria to Promote Plant Growth

Biological control using plant-growth-promoting bacteria (PGPB) has demonstrated its efficacy in controlling several plant diseases. PGPB establish close associations with plants and produce bioactive molecules and compounds able to protect plants from several aggressors [52]. PGPB improve plant growth and resistance through two mechanisms: direct mechanisms, where PGPB synthesize substances that promote plant growth, facilitate nutrient uptake, provide nitrogen, solubilize phosphorus, and sequester iron, and indirect mechanisms, where bacteria produce antibiotics, siderophores, cell-wall-degrading enzymes, ACC deaminase, and other molecules that inhibit the functioning of phytopathogenic organisms (biocontrol) [53,54,55].

Soil conditions, especially soil salinization, can significantly impact PGPB’s capacity to transform nutrients and increase plant tolerance to biotic and abiotic stresses [52,56]. In fact, several factors such as increasing salinity [57], the use of synthetic fertilizers and pesticides, and agricultural operations such as crop rotation [19] can affect PGPB’s efficiency and lead to the loss of their plant-growth-promoting (PGP) traits and their ability to sustain plant growth and resistance to harsh conditions under field conditions. Therefore, it is necessary to bioprospect unique microbial resources that can enhance plant development and tolerance in contaminated soils. Halophilic or halotolerant bacteria are microorganisms able to grow in saline environments reaching 33% NaCl [58] and under severe conditions thanks to changes in osmolarity [11,59]. Therefore, such microorganisms have the potential to establish complex interactions and modify the production of bioactive metabolites which qualify them to grow in the plant rhizosphere, and to support nutrient fluctuations, severe abiotic parameters such as salinity and pH, and biotic factors as well [60].

### 3.2. Utilization of PGPH for Sustainable Agriculture

A significant deal of interest has been generated in sustainable agriculture by the use of PGPH to enhance plant development and resilience to both biotic and abiotic diseases. Unlike PGPR, which are typically associated with plant roots in the rhizosphere, PGPH are free-living bacteria able to grow in different harsh environmental conditions such as soil, water, halophyte plant roots, etc. Certain strains of PGPH, mainly those living around halophytes, are able to colonize the roots of plants and promote plant growth [19]. Some halotolerant bacteria are entophytic and can colonize the plant’s interior tissues. For instance, Masmoudi et al. [61] proved that the halotolerant bacteria *B. velezensis* FMH2 isolated from saltwater efficiently colonize tomato roots, stems, and reach leaves’ internal tissues. In comparison to PGPB isolated from non-saline habitats, halotolerant bacteria, particularly those from the *Bacillus*, *Pseudomonas*, *Halobacillus*, and *Halomonas* genera, have demonstrated greater efficacy in improving plant growth, tolerance, and resistance in saline and non-saline conditions [9,19,62]. Halotolerant bacteria growing in harsh environmental conditions have evolved various adaptive strategies, the most important of which is the high potentiality to accumulate compatible solutes. Some species can synthetize these solutes or concentrate them from the surrounding environment. For instance, *H. elongata* is a halotolerant bacterium which takes up glycine betaine from the environment and synthesizes ectoine [63]. In addition, halotolerant bacteria can produce osmolytes, such as amino acids, sucrose, trehalose, and glycosyl glycerol, that help to maintain intracellular osmotic balance under high-salt conditions. Furthermore, most bacteria in this group can produce exopolysaccharides (EPSs), which promote the development of biofilms and increase bacteria’s resistance to a range of biotic and abiotic stresses by forming a protective layer and transforming toxic ions into non-toxic forms [64]. Moreover, several studies stated that halotolerant bacteria exhibited significant GC composition and high protein contents that decrease hydrophobicity and enhance the formation of stabilizing helix structures [65]. These properties make halotolerant bacteria particularly well-suited for enhancing plant growth in non-saline conditions and improving plant tolerance to harsh environmental conditions, including soil salinization.

### 3.3. The Role of Halobacteria in Promoting Plant Growth Facing Biotic and Abiotic Stress

Halotolerant PGPB exhibit several direct and indirect stress-related behaviors to enable the plants to cope with harsh conditions [9,11,66] (Figure 2). These include:(i)Supporting the production of non-enzymatic antioxidants such as ascorbate (ASC), glutathione (GSH), tocopherols (TCP), carotenoids (Car), and polyphenols, as well as enzymatic antioxidants such as superoxide dismutase (SOD), peroxidase (POD), catalase (CAT), and ascorbate peroxidase (APX) by plant antioxidant defense mechanisms (PPs) [66,67].(ii)Improving nutrient availability owing to their ability to fix atmospheric nitrogen, to solubilize phosphorus and potassium, as well as to produce iron-chelating siderophores [11,68].(iii)Maintaining increased stomatal conductance, boosting photosynthetic processes, and controlling ion transporter activity to improve plant selectivity, maintain the balance of the K^+^/Na^+^ ratio, prevent salt and chloride buildup, and promote nutrient uptake of both macro- and micronutrients.(iv)Generating EPS that plays a crucial part in creating a physical barrier surrounding the roots by binding Na+ cations and inhibiting their accumulation and transfer to higher plant organs in addition to protecting the bacterial cell from stressful situations [11,68,69]. EPS also promotes soil aggregation and enhances soil structure which subsequently improves water retention and plant nutrient availability [59,70].(v)Producing 1-aminocyclopropane-1-carboxylate (ACC) deaminase, the enzyme responsible for the depletion of plant ethylene levels which are increased in vegetable crops exposed to limiting conditions or pathogen attacks [71].(vi)Increasing the synthesis of phytohormones such as cytokinins, gibberellins, and auxin (primarily indole-3-acetic acid (IAA)), which affect root architecture and morphology as well as hydraulic conductivity. These root modifications provide the plant with more nutrients and greater flexibility so it can absorb the most soil water possible [72,73].(vii)Emitting volatile compounds (VOC) involved in regulation of phytohormones production and iron uptake [74], promotion of seed germination and plant growth, inducing disease resistance and abiotic stress tolerance and mediation of plant–microorganism interactions [73,75].(viii)Mediating the expression of numerous stress tolerance genes, including up-regulating genes encoding ion-transporter proteins such as malate transporter and ROS-responsive calcium channel proteins involved in cell division, ion homeostasis, and energy metabolism [15,76]. Additionally, it up-regulates the expression of genes responsible for the production of aquaporins, which induce water absorption. It also modifies the expression patterns of certain genes involved in ion homeostasis, including down-regulating the high-affinity K^+^ transporter (HKT1) and increasing sodium–hydrogen exchanger 2 (NHX2) in order to expel excess amounts of Na^+^ from cells and improve K^+^ uptake, thereby enhancing the K^+^/Na^+^ ratio when plants are exposed to salt-affected conditions [15,69,77].(ix)Protecting crops efficiently from disease attack via indirect stimulation, which is related to biocontrol. PGPH produce antimicrobial compounds, chelate the available iron in the rhizosphere to starve phytopathogens, synthesize various extracellular enzymes responsible for the hydrolysis of the fungal cell wall, efficiently colonize the niches within the rhizosphere to exclude pathogens by competing for nutrients and sites on roots, and improve “induced systemic resistance” (ISR) [78].

### 3.4. Effect of PGPH on Soil Fertility

A major problem in ensuring food security globally is maintaining soil fertility under agricultural intensification and increasing the productivity of marginal lands. The positive impact of halobacteria on soil structure and fertility has not been well described [14,75] despite the fact that the majority of studies on halotolerant/halophilic bacteria have reported their contribution to increasing plant growth and crop yield under biotic and abiotic stress factors [9,62,79]. In addition to improving plant growth and crop yield in the presence of biotic and abiotic stress factors, several halotolerant genera, including *Alcaligenes*, *Azospirillum*, *Bacillus*, *Burkholderia*, *Pseudomonas*, *Enterobacter*, and *Rhizobium,* also increase organic matter content and maintain soil composition [75] (Table 1).

Without changing the pre-existing microbial community, these bacteria increase the availability of important nutrients including nitrogen (N), phosphorous (P), potassium (K), zinc (Zn), and iron (Fe) in the inoculated soil [81,88]. The halotolerant strains of *B. safensis*, *B. pumilus*, *K. rosea*, *E. aerogenes*, and *A. veronii*, which were isolated from the rhizosphere of halophytes, were described by Mukhtar et al. [89] as effective phosphorus solubilizers that raise the availability of this element in saline soils. For instance, Ul Hassan and Bano [80] reported that *P. moraviensis* improve the P, N, and K contents of saline sodic soil by nearly 18–35%. Won et al. [90] also mentioned that the halotolerant strain *B. licheniformis* MH48 played an important role in increasing N and P contents in soil thanks to its potentialities to fix atmospheric nitrogen and to solubilize phosphate. 

The benefits of inoculating soil with halotolerant bacteria are not limited to improving the availability of macroelements (NPK); these microorganisms also help to enhance microelement contents (Fe, Mn, Zn, and Cu), to ameliorate soil structure by assimilating organic matter and to reduce pH and soil electrical conductivity (EC) [85,91]. In addition, bacterial exopolysaccharides (EPSs) play an important role in soil fertility. They form micro- and macroaggregates acting as a carbon source, ensuring the coagulation of soil particles, and resulting in the formation of humic substances, which are considered stable organic carbon forms [15,84]. EPS-forming aggregates are also key actors in increasing the active sequestration of Na+ ions, which reduce Na+ availability in degraded soils [70]. Exogenous application of PGPH also enhances dehydrogenase activity, the responsible enzyme of the biological oxidation of the organic matter present in the soil, and significantly reduces the Na^+^/K^+^ ratio, pH, EC, and uptake of exchangeable Na^+^. For example, *P. moraviensis*, *B. cereus*, and *Stenotrophomonas maltophilia* isolated from halophyte root powder and inoculated in saline–sodic ground improved soil texture, EC, pH, and organic matter contents [92]. Pankaj et al. [93] highlighted the ability of *Acinetobacter calcoaceticus*, *P. plecoglossicida*, *B. flexus*, and *B. safensis* to increase acid phosphatase, alkaline phosphatase, and dehydrogenase activities. At the same time, it reduced the Na^+^/K^+^ ratio, exchangeable Na^+^ percentage, and Na^+^ uptake rate, as well as the reduction in EC and pH. 

Moreover, some reports have shown that the use of PGPH has great potential not only for the recovery of degraded soils affected by salinity but also for restoring and maintaining lands contaminated by petroleum hydrocarbons [82,83,87], or by heavy metals (Table 1). Surfactant compounds produced by some halophilic bacteria are promising compounds that remediate polluted environments. Their amphiphilic moieties promote the compartmentalization of the hydrophobic contaminants into the internal hydrophobic cores of surfactant micelles, facilitating pollutant detachment from sediments and advocating the bioremediation processes [94,95]. Nikolopoulou et al. [96] reported that the addition of biosurfactant rhamnolipids to a solution of crude oil and sand led, after 15 days, to a degradation yield of 30% for fluorene, 20% for phenanthrene, and 10% for dibenzothiophene. 

In addition, thanks to their original properties, PGPH exhibit several mechanisms for heavy metal detoxification. Several heavy metal removal mechanisms are used, such as biosorption of heavy metals into bacterial cells, bioaccumulation, biosurfactant production, oxidation–reduction, biomineralization, transformation of toxic metal into less toxic forms, or degradation, bioleaching, and use of the heavy metal as an electron acceptor [86,97,98,99]. For example, it was reported that the strain *Bacillus firmus* TE7 detoxifies Cr (VI) by bioreduction and removes As (III) using a bio-oxidation mechanism [100]. In addition, the halotolerant strain *Shewanella loihica* PV-4 detoxifies vanadium (V (IV)) and Cr (VI) through their bioreduction into V (IV) and Cr (III) [101]. 

Thus, all these studies confirmed the important potentiality of PGPH in the recovery of soil fertility, resulting in an additional benefit of these kinds of microorganisms that includes increased plant growth, improving plant resistance to biotic and biotic stressors, and soil restoration, thereby reviving the lost vegetation and ensuring agricultural sustainability.

## 4. Interaction between PGPH and Soil Microbiota

Plant microbial communities, such as mycorrhizal fungi, are responsible for nutrient transformations and biogeochemical processes. Thus, microbiota structure is highly important in assessing soil health and quality [102]. PGPH can have a significant impact on plant microbiota. Halotolerant bacteria are well known for their ability to produce plant-growth-promoting substances, such as auxins and cytokinins, and to liberate enzymes that break down complex organic compounds into simpler forms. These substances not only promote plant growth, but can also be more readily taken up by the plant microbiota, which improve their growth and their ability to colonize plant roots [103,104]. Moreover, PGPH can help the surrounding microbial community in overcoming various stresses [102]. Halotolerant bacteria are capable of regulating the accumulation of osmolytes and other stress-related compounds, which are utilized by the plant rhizomicroflora to acquire osmoadaptation, improving growth, survival, and colonization capacity [105].

Although PGPH can have positive effects on plant microbial community, they can cause some negative impacts. The mechanisms responsible for this result are not well understood and functional explanation performance dates are still unknown [106]. However, some studies tried to find an explanation. For example, some substances produced by some PGPH may be toxic to surrounding microorganisms. PGPH, mainly non-native bacteria, can also compete with the existing community for nutritional resources, leading to total root colonization, a significant reduction in microbiota abundance or diversity, and, thereby, a disruption in the natural balance [107].

Therefore, for the commercial exploitation of halotolerant bacteria as an efficient biofertilizer, it is important to carefully select PGPH strains, examine their interaction with the native soil microbial community, and consider their potential effects on the existing microbiota before implementing their application in agriculture or other settings. These studies ensure the safe and consistent utilization of potent halotolerant bacteria on a large scale.

## 5. Limitations of Using PGPH in Agriculture

Despite the multiple beneficial effects of these bacteria on affected plants, some concerns are reported in the literature. First, most of the studies were conducted in vivo or in controlled conditions. However, when applied in the field, strains often lose their promoting effects due to the dependence of their activity on their original environments, which are different to the one in which they are inoculated [108]. Therefore, knowledge of the nature of the original environment and collecting bacteria from roots exposed to several types of stresses (alkalinity, salinity, drought, temperatures, etc.) may help to reduce variation in efficacy and withstand many of the environmental stressors to which plants are exposed [19]. Second, most of the factors influencing bacterial PGP ability and expression of biological characteristics are not well understood. Moreover, the physiological and molecular mechanisms contributing to enhanced plant growth and halotolerance remain unknown [19]. Understanding these mechanisms is critical for the best application and optimal inoculum performance. Lastly, the performance of halotolerant bacteria used in plant growth promotion should be tested over at least 2 years and under field conditions, which can combine various stresses, such as drought, salinity, and heavy metal contamination. Importantly, PGPH that are used effectively in agriculture should be studied for their efficacy in phytoremediation, biofertilization, salt stress alleviation, and biological control [79].

## 6. Recommendations for Future Research

Taking together all the information highlighted in this review, it is clear that halotolerant bacteria have the potential to play a crucial role in sustainable agriculture in the future by helping to resolve, at the same time, a range of problems related to plant growth, including abiotic stress, phytopathogenic disease, and soil fertility. However, the use of these halobacteria must be carefully managed to ensure maximum benefits. Factors such as the choice of bacterial strain, application method, doses, and environmental conditions should be well studied to avoid any impact on their effectiveness. 

The choice of appropriate bacterial strain is critical to ensure maximum benefits in sustainable agriculture. Among the most important criteria to consider is the origin of the strain (native or non-native) and its sporulation capacity. These factors determine the specific properties offered by the bacteria to improve plant growth and interaction with surrounding environmental conditions, plant species, and soil type. Nevertheless, native and sporulant halotolerant strains may be more suitable for use as a biofertilizer, since they are well adapted to local conditions, establish themselves more easily in the soil, and their spores persist in the soil for longer periods.

The choice of application method (plant irrigation, foliar spray, seeds treatments, etc.) is highly important for biofertilizer efficiency. The selection of the best method is related to plant species, size, growth stage, life cycle, and the desired effect. Conditions surrounding the plants (frequency of rainfall, temperature, and humidity) should also be considered to ensure bacteria effectiveness and persistence in the plant and soil. 

Determining the frequency, timing, and number of biofertilizer applications also plays an important role, as these parameters depend on the treated crop, the bacterial potentiality, and the environmental conditions surrounding the plant and the bacteria (species, soil type, climate, etc.). In some cases, a single application of PGPH may be sufficient to improve plant growth and resistance, while in others, repeated applications may be required to provide long-lasting benefits to the plant. Overall, the decision on what, when, how, and how often to apply PGPH depends on the specific needs of the crop and the environmental conditions. It should be based on careful evaluation of these factors and thorough analysis of all conditions that may affect biofertilizer effectiveness. 

In addition to research, it will be important to develop policies and programs that promote the use of these halotolerant bacteria in agriculture. This could include initiatives to raise awareness among farmers and policymakers about the potential benefits of these microorganisms, as well as programs to provide farmers with access to the necessary resources and training to use them effectively.

Moreover, it should be emphasized that halotolerant bacteria cannot provide a complete solution on their own and must be integrated with other sustainable agriculture strategies. The adoption of practices such as crop rotation, reduced tillage, and the use of cover crops can create a favorable environment for the growth and activity of these bacteria, leading to more sustainable agriculture in the long term. 

In other terms, to achieve sustainable agriculture with the help of halotolerant bacteria, a combination of research, policy development, and the adoption of sustainable agricultural practices is necessary. By working together towards this goal, we can harness the potential of these microorganisms to create a more sustainable and resilient food system for the future.

## 7. Conclusions and Future Directions

Halotolerant bacteria have the potential to play a crucial role in sustainable agriculture in the future. These microorganisms can help to improve crop production and support sustainable agriculture, especially in arid and semiarid situations. They have demonstrated competitive benefits over non-halotolerant strains in many habitats due to their capacity to handle challenging situations and create specialized mechanisms to survive in harsh climates. Additionally, these bacteria have the ability to improve the growth and yield of crops exposed to adverse situations such as abiotic stress and phytopathogenic disease. Despite the promising features displayed by halotolerant bacteria at the lab scale, full-scale field experiments are necessary to check their efficiency in natural environments. Thus, further studies are required on a large scale, for long periods, and in disparate conditions to evaluate the efficacy of these microbes and to unveil their survival requirements, their optimal environmental conditions, and their competitive ability with indigenous microbial populations. Therefore, to achieve sustainable agriculture with the help of halotolerant bacteria, a combination of research, policy development, and the adoption of sustainable agricultural practices can harness the potential of these microorganisms to create a more sustainable and resilient food system for the future.

## Figures and Tables

**Figure 1 microorganisms-11-01248-f001:**
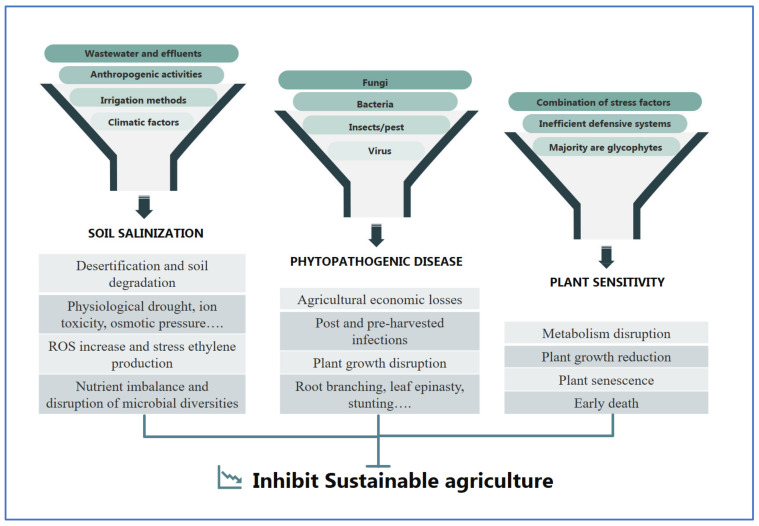
Examples of challenges limiting agriculture sustainability.

**Figure 2 microorganisms-11-01248-f002:**
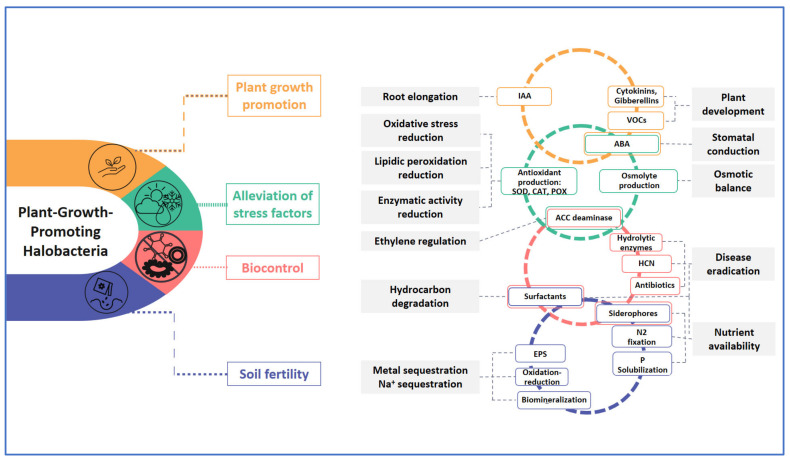
PGPH as promising tools to promote plant growth and resistance to biotic and abiotic stresses, and to sustain soil fertility.

**Table 1 microorganisms-11-01248-t001:** Examples of PGPH playing an important role in soil bioremediation.

PGPH	Role in Soil Bioremediation	Type of Affected Soil	Bioremediation Mechanisms	References
*B. cereus* *P. moraviensis* *S. maltophilia*	Improvement in P, NO_3_^−^, N, and K contents.	Saline sodic soils	Phosphate solubilization Atmospheric N_2_ fixation	[80]
*B. licheniformis* MH48	Increase in P contents	Saline soils	Phosphate solubilization	[81]
*Delftia* sp. *Achromobacter* sp. *B. kochii* AHV-KH14	Phenanthrene degradation	Oil-contaminated and saline soils	Biodegradation	[82] [83]
*P. putida* GAP-P45	Soil aggregation. Aggregate stability	Dry and saline soils	EPS production	[84]
*Pseudomonas* sp. *Thalassobacillus* sp. *Terribacillus* sp.	Decline in Na contents. Increase in Ca^2+,^ Mg^2+^ and organic matter levels.	Saline soils	Salt leaching	[85]
*K. pneumoniae* USL2S *P. putida* USL4W *P. putida* USL5W	Decrease in Hg, Pb, Cd, Ni, Cu, and Zn contents.	Acidic, heavy-metal-, and salt-contaminated soils	Bioremoval capacity	[86]
*Halobacillus* sp. EG1HP4QL	Removal of paraffins, naphthenes, mono- and bicyclic aromatic hydrocarbons, polycyclic aromatic hydrocarbons, and alcohol–benzene resins.	Oil-contaminated soils	Enzymatic activity	[87]

## Data Availability

Not applicable.

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
