# Peer review of "Halobacteria-Based Biofertilizers: A Promising Alternative for Enhancing Soil Fertility and Crop Productivity under Biotic and Abiotic Stresses—A Review"

_microorganisms, 2023, doi:10.3390/microorganisms11051248_

Round 1

Reviewer 1 Report

excellent paper
Congratulations to the authors

Just the introduction needs to be improved with some comments about the importance of the soils in the Earth Planet and our societies

Your introducion needs to show that your research is very valuable to understand the role of soils in the Earth System and the sustainability of our societies
Keesstra, S., Mol, G., de Leeuw, J., Okx, J., de Cleen, M., & Visser, S. (2018). Soil-related sustainable development goals: Four concepts to make land degradation neutrality and restoration work. Land, 7(4), 133.
Keesstra, S., Sannigrahi, S., López-Vicente, M., Pulido, M., Novara, A., Visser, S., & Kalantari, Z. (2021). The role of soils in regulation and provision of blue and green water. Philosophical Transactions of the Royal Society B, 376(1834), 20200175.
Keesstra, S., Nunes, J., Novara, A., Finger, D., Avelar, D., Kalantari, Z., & Cerdà, A. (2018). The superior effect of nature based solutions in land management for enhancing ecosystem services. Science of the Total Environment, 610, 997-1009.

Author Response

Reviewer #1:

  1. Just the introduction needs to be improved with some comments about the importance of the soils in the Earth Planet and our societies

Your introduction needs to show that your research is very valuable to understand the role of soils in the Earth System and the sustainability of our societies

Keesstra, S., Mol, G., de Leeuw, J., Okx, J., de Cleen, M., & Visser, S. (2018). Soil-related sustainable development goals: Four concepts to make land degradation neutrality and restoration work. Land, 7(4), 133.

Keesstra, S., Sannigrahi, S., López-Vicente, M., Pulido, M., Novara, A., Visser, S., & Kalantari, Z. (2021). The role of soils in regulation and provision of blue and green water. Philosophical Transactions of the Royal Society B, 376(1834), 20200175.

Keesstra, S., Nunes, J., Novara, A., Finger, D., Avelar, D., Kalantari, Z., & Cerdà, A. (2018). The superior effect of nature based solutions in land management for enhancing ecosystem services. Science of the Total Environment, 610, 997-1009.

Response:

- Thank you so much for your suggestion. As recommended by the reviewer, the introduction has been improved and in the introduction of the revised manuscript Page 1-2, L45-58, a paragraph describing the negative effects of intensive agricultural practices on soil has been added as follow:  

“However, intensive agricultural practices have various negative effects on soil. It is the main reason of physical degradation caused by agricultural machinery, drainage and dewatering. The overuse of manure and fertilizers can lead to organic and inorganic pollution, which can result in soil salinization, acidification and chemical degradation. Moreover, synthetic fertilizers, pesticides, and monoculture can negatively affect soil biodiversity and ecosystem functioning leading to a biological degradation of soil or-ganic matter, which is essential for soil structure, water-holding capacity, and nutrient cycling [3].

Additionally, intensive agriculture is responsible for approximately 92% of fresh-water consumption, raising several concerns about water resources uses [4]. Irrigation with saline water accentuates soil salinization process and affects yields production.

In other words, a growing demand for agricultural products has been observed while the amount of cultivated land, the soil structure and natural resources availabil-ity has been steadily decreasing [5-8].

- In the reference section, three new references has been added as follow:

  1. Keesstra, S.; Mol, G.; De Leeuw, J.; Okx, J.; Molenaar, C.; De Cleen, M.; Visser, S. Soil-Related Sustainable Development Goals: Four Concepts to Make Land Degradation Neutrality and Restoration Work. Land 2018, 7, 133, doi:10.3390/land7040133.
  2. Keesstra, S.; Sannigrahi, S.; López-Vicente, M.; Pulido, M.; Novara, A.; Visser, S.; Kalantari, Z. The Role of Soils in Regulation and Provision of Blue and Green Water. Philosophical Transactions of the Royal Society B: Biological Sciences 2021, 376, 20200175, doi:10.1098/rstb.2020.0175.
  3. Keesstra, S.; Nunes, J.; Novara, A.; Finger, D.; Avelar, D.; Kalantari, Z.; Cerdà, A. The Superior Effect of Nature Based Solutions in Land Management for Enhancing Ecosystem Services. Science of The Total Environment 2018, 610–611, 997–1009, doi:10.1016/j.scitotenv.2017.08.077.

Reviewer 2 Report

Dear authors

Is it known how PGPH bacteria with mycorrhizal associations interact with fungi? Normally, bacteria and fungi do not like each other because they occupy similar ecological niches and try to restrict each other by producing volatile substances - secondary metabolites (Forests 2018, 9, 597; doi:10.3390/f9100597). This then leads to the disappearance of mycorrhizae and even the death of some fine roots. 

It is also unclear whether the PGPH bacteria are endophytic and colonise the leaves or the soil zone of the rhizosphere, and whether they are plant-specific or not? If so, is a single application of these bacteria likely to be sufficient? If not, when and how often should they be applied, or every year? Can environmental conditions with too little moisture in the soil (drought) cause the bacteria to die? Conversely, can excessive rainfall wash the bacteria out of the soil, and in either case do the applications need to be repeated? Is it known how the mass application of PGPH bacteria in agriculture affects soil biodiversity? Are the selected PGPH bacteria of foreign origin and could be invasive? I propose to address these questions in the discussion.

 In Figure 1, the white lettering on the green background is only faintly visible and "PHYTOPATHOGENIC DISEASE is partially obscured. In Figure 2, the lettering on the right side on the grey background is of quite poor quality.

L64 Is the article really the first review article on this topic? I would also suggest reviewing and possibly including the recent literature on the mitigation of salt stress by site-adapted symbioses published in Forests 2022, https://doi.org/10.3390/f13040586.

There is a bit of repetition of L 67 fungal attack abiotic stressor and L76-77 etc.

English is fine

Author Response

Reviewer #:2

Comment 1

Is it known how PGPH bacteria with mycorrhizal associations interact with fungi? Normally, bacteria and fungi do not like each other because they occupy similar ecological niches and try to restrict each other by producing volatile substances - secondary metabolites (Forests 2018, 9, 597; doi:10.3390/f9100597). This then leads to the disappearance of mycorrhizae and even the death of some fine roots.

Response 1

Thank you for your valuable remark. It is true that some plant growth promoting halobacteria may have some negative impacts on the surrounding plant microbial community. However, many strains of PGPH were described for their  positive impacts on the existing microbiota such as mycorrhizae and their abundance mainly in presence of a salt stress condition. Actually, we are writing a review, which will describe the interactions between PGPH and soil microbiota and the possibility to combine between PGPH and mycorrhizae to improve plant resistance to biotic and abiotic stresses. However, in the current manuscript a short paragraph has been added P10, L421-446 to describe some interaction mechanisms as follow:

5. Interaction between PGPH and soil microbiota

Plant microbial communities, such as mycorrhizal fungi, are responsible for nutrient transformations and biogeochemical processes. Thus, microbiota structure is highly important to assess soil health and quality [104]. PGPH can have a significant impact on plant microbiota. Halotolerant bacteria are well known by their ability to produce plant growth promoting substances, such as auxins and cytokinins, and to liberate enzymes that break down complex organic compounds into simpler forms. These substances not only promote plant growth, but can also be more readily taken up by the plant micro-biota, which improve their growth and their ability to colonize plant roots [104,105]. Moreover, PGPH can help the surrounding microbial community in overcoming various stresses [106]. Halotolerant bacteria are capable of regulating the accumulation of osmo-lytes and other stress-related compounds, which are utilized by the plant rhizo-microflora to acquire osmoadaptation, improving growth, survival, and coloniza-tion capacity [107].

Although PGPH can have positive effects on plant microbial community, they can cause some negative impacts. The mechanisms responsible for this result are not well understood and functional explanation performance date still unknown [108]. However, some studies tried to find explanation. For example, some substances produced by some PGPH may be toxic to surrounding microorganisms. PGPH, mainly non-native bacteria, can also compete existing community to nutritional resources leading to a to-tal root colonization, a significant reduction in microbiota abundance or diversity, and thereby a disruption in the natural balance [109].

 Therefore, for a commercial exploitation of halotolerant bacteria as an efficient biofertilizer, it is important to carefully select PGPH strains, examine their interaction with native soil microbial community, and consider their potential effects on the ex-isting microbiota before their application in agriculture or other settings. These studies ensure a safe and consistent utilization of potent halotolerant bacteria at large scale.

The following references has been added to the reference section

  1. Di Salvo, L.P.; Cellucci, G.C.; Carlino, M.E.; García de Salamone, I.E. Plant Growth-Promoting Rhizobacteria Inoculation and Nitrogen Fertilization Increase Maize (Zea Mays L.) Grain Yield and Modified Rhizosphere Microbial Communities. Applied Soil Ecology 2018, 126, 113–120, doi:10.1016/j.apsoil.2018.02.010.
  2. Khalid, M.; Hassani, D.; Bilal, M.; Asad, F.; Huang, D. Influence of Bio-Fertilizer Containing Beneficial Fungi and Rhizospheric Bacteria on Health Promoting Compounds and Antioxidant Activity of Spinacia Oleracea L. Botanical Studies 2017, 58, 35, doi:10.1186/s40529-017-0189-3.
  3. Tirry, N.; Kouchou, A.; Laghmari, G.; Lemjereb, M.; Hnadi, H.; Amrani, K.; Bahafid, W.; El Ghachtouli, N. Improved Salinity Tolerance of Medicago Sativa and Soil Enzyme Activities by PGPR. Biocatalysis and Agricultural Biotechnology 2021, 31, 101914, doi:10.1016/j.bcab.2021.101914.
  4. Chandra, A.; Singh, M. Biosynthesis of Amino Acid Functionalized Silver Nanoparticles for Potential Catalytic and Oxygen Sensing Applications. Inorg. Chem. Front. 2018, 5, 233–257, doi:10.1039/C7QI00569E.
  5. Bzdyk, R.M.; Olchowik, J.; Studnicki, M.; Oszako, T.; Sikora, K.; Szmidla, H.; Hilszczańska, D. The Impact of Effective Microorganisms (EM) and Organic and Mineral Fertilizers on the Growth and Mycorrhizal Colonization of Fagus Sylvatica and Quercus Robur Seedlings in a Bare-Root Nursery Experiment. Forests 2018, 9, 597, doi:10.3390/f9100597.
  6. Deng, S.; Wipf, H.M.-L.; Pierroz, G.; Raab, T.K.; Khanna, R.; Coleman-Derr, D. A Plant Growth-Promoting Microbial Soil Amendment Dynamically Alters the Strawberry Root Bacterial Microbiome. Sci Rep 2019, 9, 17677, doi:10.1038/s41598-019-53623-2.

Comment 2

It is also unclear whether the PGPH bacteria are endophytic and colonise the leaves or the soil zone of the rhizosphere, and whether they are plant-specific or not?

Response 2

Thank you for your question. Some PGPH bacteria can be epiphytic and endophytic. These microorganisms are able to colonize all internal plant tissues: root, stem and leaves to ensure a better promotion of plant growth. In the revised manuscript P6, L282 a paragraph has been added as follow:

“Unlike PGPR, which are typically associated with plant roots in the rhizosphere, PGPH are free-living bacteria able to grow in different harsh environmental conditions such as soil, water, halophyte plant roots, etc. Certain strains of PGPH, mainly those living surrounding halophytes, can be able to colonize the plant roots of plants and promote plant growth [61]. Some halotolerant bacteria are entophytic and can colonize the plant's interior tissues. For instance, Masmoudi et al. [62] proved that the halotolerant bacteria B. velezensis FMH2 isolated from solartaltern colonize efficiently tomato roots, stem and reach leave’s internal tissues.”

The following references has been added to the reference section:

  1. Etesami, H.; Glick, B.R. Halotolerant Plant Growth–Promoting Bacteria: Prospects for Alleviating Salinity Stress in Plants. Environmental and Experimental Botany 2020, 178, 104124, doi:10.1016/j.envexpbot.2020.104124.
  2. Masmoudi, F.; Tounsi, S.; Dunlap, C.A.; Trigui, M. Endophytic Halotolerant Bacillus Velezensis FMH2 Alleviates Salt Stress on Tomato Plants by Improving Plant Growth and Altering Physiological and Antioxidant Responses. Plant Physiology and Biochemistry 2021, 165, 217–227, doi:10.1016/j.plaphy.2021.05.025.

Comment 3

If so, is a single application of these bacteria likely to be sufficient? If not, when and how often should they be applied, or every year?  Can environmental conditions with too little moisture in the soil (drought) cause the bacteria to die? Conversely, can excessive rainfall wash the bacteria out of the soil, and in either case do the applications need to be repeated? Is it known how the mass application of PGPH bacteria in agriculture affects soil biodiversity? Are the selected PGPH bacteria of foreign origin and could be invasive? I propose to address these questions in the discussion.

Response 3

Thank you so much for your questions. The use of PGPH as biofertilizer must be carefully managed to ensure maximum benefits. Factors such as the choice of bacterial strain, application method, doses, and environmental conditions should be well studied to avoid any impact on their effectiveness.

  • The choice of appropriate bacterial depend on: the specific needs of the crop, properties offered by the bacteria to improve plant growth and its interaction with the plant species, soil type, and environmental conditions. Native and sporulant halotolerant strains may be more adequate for use as a biofertilizer, since they are already adapted to the local conditions, generally they are invasiveness and do not negatively impact the local ecosystem and spores have the advantage of being more resistant to environmental stresses and easier in formulation and storage.

  • The choice of application method (plant irrigation, foliar spray, seeds treatments, etc.) is highly important for biofertilizer efficiency. The selection of the best method is related to plant species, size, growth stage, life cycle, and the desired effect. Conditions surrounding the plants (frequency of rainfall, temperature, and humidity) should be considered to ensure bacteria effectiveness and persistence in the plant and soil.

  • The frequency, timing and number of PGPH applications depends on a variety of parameters, including the bacterium strain, the treated crop, and the environmental conditions. In some cases, a single application of PGPH may be sufficient, while in other cases, repeated applications may be required. The frequency and timing of PGPH application could also be related to the plant cycle (the growth stage, flowers, fruits), as well as the crop's life period (annual/perennial).

In summary, the effectiveness of halotolerant bacteria and the frequency of application depend on several factors, and careful consideration should be taken to ensure the safety and effectiveness of their use in agriculture.

In the revised manuscript the paragraph P11-12, L 471-495 has been rephrased as follow “Factors such as the choice of bacterial strain, application method, doses, and environmental conditions should be well studied to avoid any impact on their effectiveness.

The choice of appropriate bacterial strain is critical to ensure maximum benefits in sustainable agriculture. Among the most important criteria to consider the origin of the strain (native or non-native) and its sporulation capacity. These factors determine the specific properties offered by the bacteria to improve plant growth and interaction with surrounding environmental conditions, plant species, and soil type. Nevertheless, native and sporulant halotolerant strains may be more suitable for use as a biofertilizer, since they are well adapted to local conditions, establish themselves more easily in the soil, and their spores persist in the soil for longer periods.

The choice of application method (plant irrigation, foliar spray, seeds treatments, etc.) is highly important for biofertilizer efficiency. The selection of the best method is related to plant species, size, growth stage, life cycle, and the desired effect. Conditions surrounding the plants (frequency of rainfall, temperature, and humidity) should also be considered to ensure bacteria  effectiveness and persistence in the plant and soil.

Determining the frequency, timing, and number of biofertilizer applications also plays an important role, as these parameters depend on the treated crop, the bacterial potentiality and the environmental conditions surrounding the plant and the bacteria (species, soil type, climate, etc.). In some cases, a single application of PGPH may be sufficient to improve plant growth and resistance, while in others, repeated applica-tions may be required to provide long-lasting benefits to the plant.

Overall, the deci-sion on what, when how, and how often to apply PGPH depends on the specific needs of the crop and the environmental conditions. It should be based on careful evaluation of these factors and thorough analysis of all conditions that may affect bio-fertilizer effectiveness.

Comment 4

 In Figure 1, the white lettering on the green background is only faintly visible and "PHYTOPATHOGENIC DISEASE is partially obscured. In Figure 2, the lettering on the right side on the grey background is of quite poor quality.

Response 4:

Thank you very much for your suggestion. Colors of the background in the figure 1 have been changed to light green and white lettering have been changed to black lettering. In Figure 2, the lettering on the right side on the grey background has been written in bigger size.

Comment 5:

L64 Is the article really the first review article on this topic? I would also suggest reviewing and possibly including the recent literature on the mitigation of salt stress by site-adapted symbioses published in Forests 2022, https://doi.org/10.3390/f13040586.

Response 5

Thank you for your question. While numerous reports and reviews have highlighted the beneficial effects of halotolerant bacteria in plant growth promotion and mitigating plant salt stress, most of these studies have focused solely on this aspect. Only a few reports have investigated the ability of PGPH to play other roles, such as, biocontrol, and soil remediation. Interestingly, to the best of our knowledge, no previous review article that explores the multifaceted potential of PGPH in a single paper and our review aims to fill this gap by examining the multitasking ability of these strains and highlighting their various roles in plant growth promotion, biocontrol, soil remediation, and stress alleviation.

To avoid any confusion, in the revised manuscript the whole paragraph P2, L70-L86 has been rephrased as follow:

“Several previous reviews highlighted the important role played by PGPH in plant growth promotion and mitigating plant drought and salt stresses, and most of these studies have focused solely on this aspect. [11–14]. Only a few reports focused on the PGPH potential to play other roles such as biocontrol, and recovery of contaminated soils [14,15].

Interestingly, to the best of our knowledge, no previous review article summarize in a single paper the various roles played by PGPH in protecting and increasing crop productivity under biotic and biotic stresses, recovering degraded lands, and sustaining agricultural products. Our review article is the first exploring the multitasking ability of halotolerant bacteria isolated from saline environments and halophyte rhi-zospheres, which contribute not only in alleviating abiotic stresses but also in improving plant protection against biotic diseases, promoting plant growth, and aiding in the recovery of affected soils. It also discusses the challenges facing conventional agriculture, particularly biotic (fungal attack) and abiotic stresses (salt stress, drought stress, etc.). The review highlights also the direct and indirect pathways used by PGPH as bio-inoculants to improve crop tolerance to abiotic stresses and fight against phytopathogenic attacks. Finally, it highlights PGPH limits in agriculture and presents authors recommendation for future research and application of halotolerant bacteria as biofertilizers.”

In the reference section the following reference has been added as follow:

  1. Rabhi, N.E.H.; Cherif-Silini, H.; Silini, A.; Alenezi, F.N.; Chenari Bouket, A.; Oszako, T.; Belbahri, L. Alleviation of Salt Stress via Habitat-Adapted Symbiosis. Forests 2022, 13, 586, doi:10.3390/f13040586”.

Comment 6:

There is a bit of repetition of L 67 fungal attack abiotic stressor and L76-77 etc.

Response 6:

Thank you very much for your remarks, Your remark has been taken under consideration and now in the revised manuscript ‘Fungal attack’ is mentioned only one time L82 and abiotic stressor has been removed and did not exist anymore in the revised manuscript.

Reviewer 3 Report

Review for

Review

 Halobacteria-based Biofertilizers: A Promising Alternative For

 Enhancing Soil Fertility and Crop Productivity Under Biotic

 and Abiotic Stresses. A review

by

Fatma Masmoudi, Mohammed AL-Safran, Hareb AL Jabri, Hoda Hosseini, Mohammed Trigui, Sami Sayadi, Slim Tounsi and Imen Saadaoui

Abiotic and biotic stresses such as salt stress and fungal infections significantly affect plant

 growth and productivity, leading to reduced crop yield. This review is then most welcome.

-------------------

Traditional methods of managing stress  factors, such as developing resistant varieties, chemical fertilizers, and pesticides, have shown limited success.

to my opinion, this is absolutely not true

huge progress of agricultural production on the planet, based on developing resistant varieties, chemical fertilizers, and pesticides

the problem is more on pollution by pesticides, health effects on these

we should now avoid most of them

----------------------

Halotolerant bacteria found in saline environments have potential as plant promoters

 under stressful conditions. These microorganisms produce bioactive molecules and plant growth

 regulators, making them a promising agent for enhancing soil fertility, improving plant resistance

 to adversities, and increasing crop production

fully true but the main question is about true application of these halotolerant bacteria in NON-saline environments

------------------------------

This review highlights the capability of plant growth

promoting halo-bacteria (PGPH) to stimulate plant growth, strengthen plant tolerance and resistance to biotic and abiotic stressors, and sustain soil fertility

----------------------

original figure 1?? taken from??

-----------------------

5. Limitations of using PGPH in agriculture

 Although the multiple beneficial effects these bacteria on affected plants, some concerns are reported in literature

what about real life?  true applications already existing for large farms? large crops?

same comment for N-fixing microorganisms? decades of research, true applications?

--------------------

6. Recommendations for future research

nice part, very objective

Author Response

Reviewer #:3

Comment 1

Traditional methods of managing stress factors, such as developing resistant varieties, chemical fertilizers, and pesticides, have shown limited success. To my opinion, this is absolutely not true huge progress of agricultural production on the planet, based on developing resistant varieties, chemical fertilizers, and pesticides. The problem is more on pollution by pesticides, health effects on these we should now avoid most of them.

Response 1

Thank you for your remark. I completely agree with you that resistant varieties, chemical fertilizers, and pesticides have contributed to a significant increase in agricultural production over the years.

Here in our manuscript when we stated, “Abiotic and biotic stresses such as salt stress and fungal infections significantly affect plant growth and productivity, leading to reduced crop yield. Traditional methods of managing stress factors, such as developing resistant varieties, chemical fertilizers, and pesticides, have shown limited success.” We were referring to their limited success in managing both abiotic and biotic stress factors. For example, most of resistant varieties can resist either biotic or abiotic stress factors and it is not evident whether generated resistant varieties can resist to combined stress factors (salt stress and fungal disease for example). Similarly, while chemical fertilizers, and pesticides are very efficient to promote plant growth and eradicating phytopathogenic disease, they seem to be inefficient to promote plant resistance when abiotic stress factor are present.

To avoid any confusion and to clarify this point, the sentence in the revised manuscript, Abstract section,P1, L23-25 of has been rephrased as follow:

“Traditional methods of managing stress factors, such as developing resistant varieties, chemical fertilizers, and pesticides, have shown limited success in the presence of combined biotic and abiotic stress factors”

Comment 2

Halotolerant bacteria found in saline environments have potential as plant promoters under stressful conditions. These microorganisms produce bioactive molecules and plant growth regulators, making them a promising agent for enhancing soil fertility, improving plant resistance to adversities, and increasing crop production

Fully true but the main question is about true application of these halotolerant bacteria in NON-saline environments

Response 2

Thank you for your question and remark. Halotolerant bacteria are microorganisms able to grow in presence as well as in absence of salt. As reported in this review article, PGPH are able to promote plant growth in presence as well as in absence of saline condition. Several researches has shown that some halotolerant bacteria can improve plant growth and productivity under non-saline conditions and this is their strength. For instance, Masmoudi et al, 2021 (DOI https://doi.org/10.1007/s00299-021-02702-8) reported the ability of the halotolerant strain B. velezensis FMH45 to produce PGP traits in vitro and promote tomato plant growth in vivo in both saline and non-saline condition. Furthermore, PGPH are known for their biocontrol efficiency. In most cases, plant infection occur in non-saline conditions because phytopathogens are not able to thrive in presence of salt stress, which proves the efficiency of PGPH  in non-saline conditions.

However, more researches are required to understand their potential in non-saline environments and to optimize their use in different agricultural systems.

To avoid any kind of confusion and misunderstanding the sentence “in non-saline conditions” has been added in the revised manuscript as follow:

-L30: This review highlights the capability of plant growth promoting halo-bacteria (PGPH) to stimulate plant growth in non-saline conditions, strengthen plant tolerance and resistance to biotic and abiotic stressors, and sustain soil fertility.

-L64 :Plant growth promoting halobacteria (PGPH) have been found to be highly effective in promoting plant growth in non-saline conditions and improving soil health, particu-larly in saline and arid environments [7].

-L284: In comparison to PGPB isolated from non-saline habitats, halotolerant bacteria, particularly those from the Bacillus, Pseudomonas, Halobacillus, and Halomonas genera, have demonstrated greater efficacy in improving plant growth in non-saline conditions, tol-erance and resistance to stress factors [[7]6,15].

-L299: These properties make halotolerant bacteria particularly well-suited for enhancing plant growth in non-saline conditions and improving plant tolerance to harsh envi-ronmental conditions, including soil salinization

In the reference section, a new reference has been added as follow:

  1. Masmoudi, F.; Tounsi, S.; Dunlap, C.A.; Trigui, M. Halotolerant Bacillus Spizizenii FMH45 Promoting Growth, Physiological, and Antioxidant Parameters of Tomato Plants Exposed to Salt Stress. Plant Cell Rep 2021, 40, 1199–1213, doi:10.1007/s00299-021-02702-8.

Comment 3

original figure 1?? taken from??

Response

Thank you for your question. “Figure 1. Examples of challenges limiting agriculture sustainability” was not taken from a site or scientific paper. It was generated thanks to the efforts of the authors to summarize the most important challenges facing agriculture sustainability.

Comment 4

  1. Limitations of using PGPH in agriculture

Although the multiple beneficial effects these bacteria on affected plants, some concerns are reported in literature what about real life?  true applications already existing for large farms? large crops? same comment for N-fixing microorganisms? decades of research, true applications

Response 4

Thank you for you remarks and questions. I completely agree with you that while there have been promising results in laboratory and in vivo-trials, practical applications in large-scale farming are still limited. In our manuscript, we have highlighted this point and tried to explain reasons that limit the large-scale application such as bacterial efficiency reduction, economic cost, combinations of stress factors, etc.  As researchers, we are currently trying to apply promising halotolerant strains on a large scale farming in different agricultural systems (soil cultivation, hydroponic cultivation) and under different stress factors. Promising results has been obtained and research papers will be published in the near future.

Round 2

Reviewer 2 Report

Dear authors, thank you for responding to all my comments, the scholarly discussion and the justifications supported by the published literature. I have no further questions and congratulate you on your interesting work

the work is written in clear English, to which I have no objections